# Changes in neural progenitor lineage composition during astrocytic differentiation of human iPSCs

Zongze Li[1,2], Lucia Fernandez Cardo[1], Michal Rokicki[1], Jimena Monzón-Sandoval[1], Viola Volpato[1], Frank Wessely[1], Caleb Webber[1]*, Meng Li[2]*

[1]Dementia Research Institute, School of Medicine, Cardiff University, Cardiff, United Kingdom; [2]Neuroscience and Mental Health Innovation Institute, School of Medicine, Cardiff University, Cardiff, United Kingdom

## eLife Assessment

The manuscript by Li and coworkers analyzed astrocytic differentiation of midbrain floor plate-patterned neural cells originating from human iPS cells, with a LMX1A reporter. This **valuable** work identifies transcriptomic differences at the single-cell level, between astrocytes generated from LMX1A reporter positive or negative cells, as well as non-patterned astrocytes and neurons. The evidence is **solid**, but the paper can be strengthened by further analyses of the transcriptomic data, and astrocytic morphology; also, searching for some of the differentially expressed genes by immunohistochemistry in different regions of the mammalian brain, or in human specimens, would be very informative.

*For correspondence:
webberc4@cardiff.ac.uk (CW);
lim26@cardiff.ac.uk (ML)

Competing interest: The authors declare that no competing interests exist.

**Abstract** The regional specificity of stem cell-derived astrocytes is believed to be an important prerequisite for their application in disease modelling and cell-based therapies. Due to the lack of subtype-defining markers for astrocytes in different regions of the brain, the regional identity of in vitro-derived astrocytes is often declared by the dominant positional characteristics of their antecedent neural progenitors, patterned to a fate of interest, with the assumption that the positional trait is preserved by the derived astrocytes via linear descent. Using a human induced pluripotent stem cell line designed for tracing derivatives of LMX1A⁺ cells combined with a ventral midbrain induction paradigm, we show that astrocytes originating from LMX1A⁺ progenitors can only be generated if these progenitors are purified prior to the astrocyte differentiation process, or their progenies are gradually lost to progenies of LMX1A⁻ progenitors. This finding indicates that the lineage composition of iPSC-derived astrocytes may not accurately recapitulate the founder progenitor population. Using deep single-cell RNA sequencing, we identified distinct transcriptomic signatures in astrocytes derived from the LMX1A⁺ progenitor cells. Our study highlights the need for rigorous characterization of pluripotent stem cell-derived regional astrocytes and provides a resource for assessing LMX1A⁺ ventral midbrain progenitor-derived human astrocytes.

## Introduction

Astrocytes are the most abundant cell type in the brain. They play important roles in the central nervous system in supporting neuronal survival and synaptic activities, including regulation of ionic homeostasis, providing energetic support, elimination of oxidative stress, and neurotransmitter removal and recycling (*Verkhratsky and Nedergaard, 2018*). Abnormalities in astrocytes have been linked to various neurodegenerative and neurodevelopmental disorders such as Parkinson's disease,

Alzheimer's disease, Huntington's disease, autism spectrum disorders, and Alexander's disease (*Molofsky et al., 2012*; *Phatnani and Maniatis, 2015*; *Booth et al., 2017*). Therefore, there is a growing interest in using human pluripotent stem cell (PSC)-derived astrocytes for in vitro disease modeling (*Chandrasekaran et al., 2016*).

Contrary to the widely held belief that astrocytes in the brain are largely identical, recent studies have revealed a diversity in their transcriptomic profiles, physiological properties, and functions (*Oberheim et al., 2009*; *Schober et al., 2022*). Single-cell and spatial transcriptomic studies have identified several astrocyte subpopulations in the mouse cortex (*Zhu et al., 2018*; *Batiuk et al., 2020*; *Bayraktar et al., 2020*). In humans, although astrocyte heterogeneity remains largely elusive, heterogeneity in radial glia across brain regions and within the midbrain has been reported (*La Manno et al., 2016*; *Li et al., 2023*). Furthermore, different molecular and physiological features as well as distinct responses to stimuli have been observed in astrocytes from different mouse brain regions (*Takata and Hirase, 2008*; *Chai et al., 2017*; *Morel et al., 2017*; *Itoh et al., 2018*; *Kostuk et al., 2019*; *Makarava et al., 2019*; *Xin et al., 2019*; *Lozzi et al., 2020*). Indeed, astrocyte heterogeneity has been suggested to underlie the regional susceptibility to human diseases (*Schober et al., 2022*). Therefore, recapitulating astrocyte regional specificity in PSC-derived astrocytes is generally accepted as an important prerequisite.

Several studies have described the generation of regional astrocytes from human embryonic stem cells or induced pluripotent stem cells (iPSCs), including the forebrain (*Krencik et al., 2011*; *Zhou et al., 2016*; *Tcw et al., 2017*; *Lin et al., 2018*; *Bradley et al., 2019*; *Hedegaard et al., 2020*; *Peteri et al., 2021*), and ventral midbrain (*Booth et al., 2019*; *Barbuti et al., 2020*; *Crompton et al., 2021*; *de Rus Jacquet et al., 2021*), hindbrain and spinal cord (*Roybon et al., 2013*; *Serio et al., 2013*; *Holmqvist et al., 2015*; *Bradley et al., 2019*; *di Domenico et al., 2019*; *Yun et al., 2019*). The regional identity of these astrocytes is typically evaluated at the stage of early neural progenitors generated via cell-type- or region-directed neural patterning protocols, with the assumption that the positional characteristics will be faithfully preserved in the final astrocyte products. However, astrocyte production in vitro involves an extended period of astrocytic fate induction and progenitor expansion using FGF and EGF, and substantial literature has reported alterations in region-specific gene expression and/or neurogenic competence in expanded neural progenitors (*Jain et al., 2003*; *Sun et al., 2008*; *Koch et al., 2009*; *Falk et al., 2012*). Therefore, better characterization of PSC-derived astrocytes and their lineage-specific features is needed to advance our knowledge of the molecular heterogeneity of human astrocytes.

Using a human iPSC line that allows the tracing of LMX1A-expressing ventral midbrain neural progenitors and their differentiated progeny (*Cardo et al., 2023*), we discovered an unexpected gradual depletion of LMX1A⁺ progenitor progeny during astrocyte induction from a bulk population of ventral midbrain patterned progenitors, despite LMX1A⁺ progenitors being the predominant starting population. However, LMX1A⁺ progenitor-derived astrocytes can be generated if astrocytic induction is initiated from purified LMX1A⁺ progenitors, indicating that the positional constituents of the founding cell population may not be preserved faithfully in the derived astrocytes. Single-cell RNA sequencing (scRNAseq) of astrocytes derived from both parental populations identified distinct transcriptomic signatures, providing a useful resource for the assessment of a defined lineage of human PSC-derived midbrain astrocytes.

## Results

### Depletion of LMX1A⁺ progenitors and/or derivatives in ventral midbrain patterned neural progenitor cultures during astrogenic induction

To investigate whether regionally patterned neural progenitors retain their lineage identity during astrogenic induction and glial progenitor expansion, we made use of the LMX1A-Cre/AAVS1-BFP iPSCs tracer line, in which LMX1A-driven Cre activates BFP expression, thereby enabling tracking of LMX1A⁺ ventral midbrain progenitors and their differentiated progeny (*Cardo et al., 2023*). We differentiated the LMX1A-Cre/AAVS1-BFP iPSCs towards the ventral midbrain fate following a modified protocol based on Jaeger et al. and Kriks et al (*Jaeger et al., 2011*; *Krencik et al., 2011*; *Figure 1A*). Immunocytochemistry of day (d) 19 cultures confirmed a high proportion of cells expressing BFP

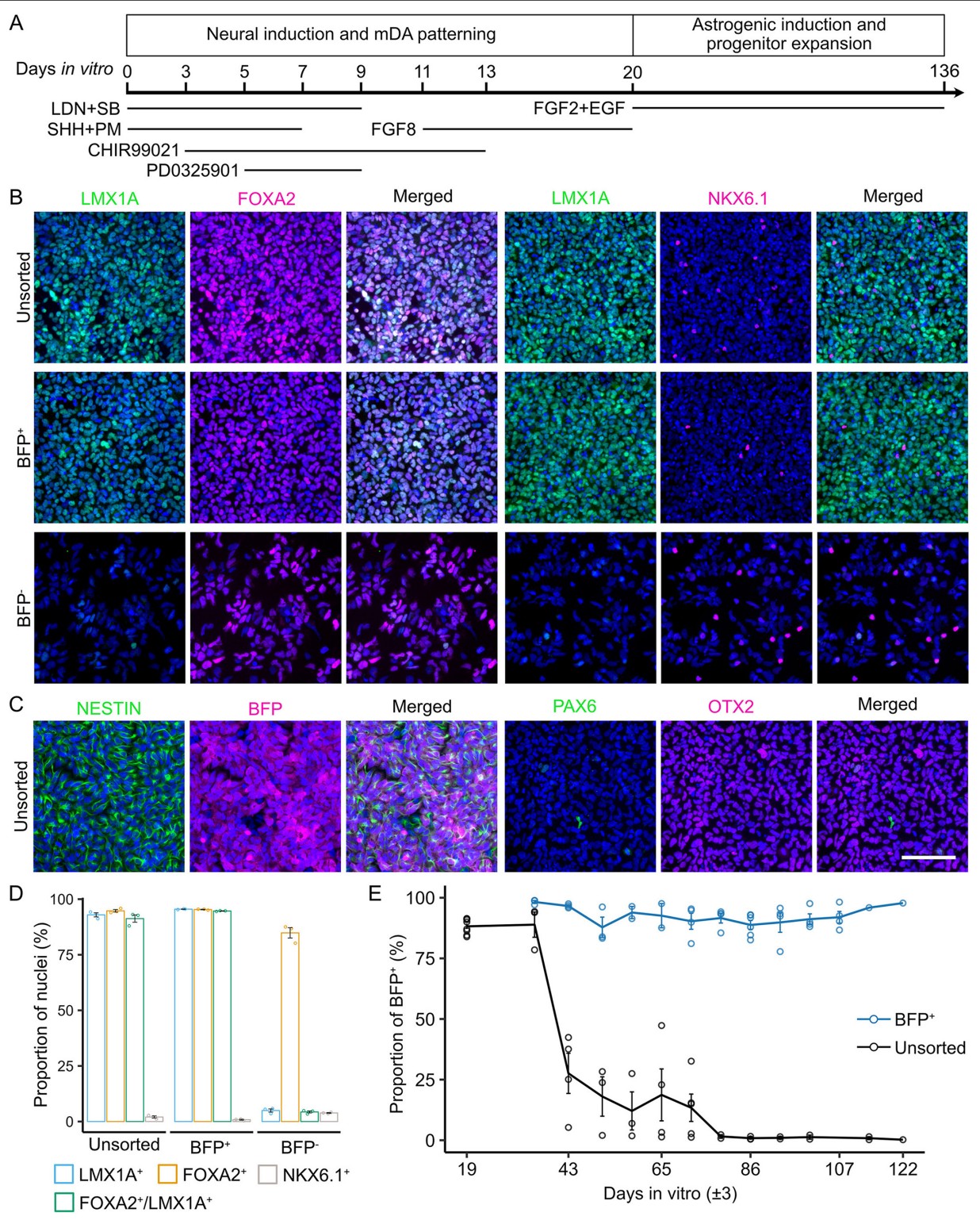

**Figure 1.** Depletion of LMX1A+ progenitors and their derivatives during astrogenic induction in ventral midbrain patterned neural progenitor cultures. (**A**) Schematic diagram of ventral midbrain neural differentiation and astrogenic induction. (**B, C**) Representative view of immunocytochemistry of ventral midbrain neural progenitor markers and other regional markers in d19 unsorted, sorted BFP+, and sorted BFP- population. Scale bar represents 100 μm. Images shown were cropped to 300 μm×300 μm by randomly selecting the region of interest in the nuclei-only channel (uncropped grayscale images are shown in *Figure 1—figure supplement 1A*). (**D**) Quantification of marker expression in unsorted, sorted BFP+, and sorted BFP- population. Error bars represent the standard error of means (SEM) of three independent experiments. (**E**) Flow cytometry quantification of unsorted and BFP+ population

*Figure 1 continued on next page*

*Figure 1 continued*

during astrogenic induction and progenitor expansion. Each data point represents one biological replicate. The gating strategy used is shown in *Figure 1—figure supplement 1B*.

The online version of this article includes the following figure supplement(s) for figure 1:

**Figure supplement 1.** Original images of immunocytochemistry of d19 progenitors and gating strategy of BFP flow cytometry analysis.

(96.01±0.42%) and ventral midbrain progenitor markers LMX1A (92.94±0.91%), FOXA2 (94.76±0.57%), and OTX2 (97.82±0.28%; *Figure 1B–D* with the original images shown in *Figure 1—figure supplement 1A*). Most cells (91.26±1.64%) co-expressed LMX1A and FOXA2 (*Figure 1B and D*). At this stage, all BFP$^+$ cells stained positive for the pan-neural progenitor marker NESTIN (*Figure 1C*). We detected a small proportion of cells expressing the midbrain basal plate marker NKX6.1 (2.03±0.47%, *Figure 1B and C*), whose expression domain in the early developing ventral midbrain partially overlap with that of LMX1A (*Andersson et al., 2006*). However, few PAX6$^+$ cells were present (*Figure 1C*), which marks the forebrain and lateral midbrain (*Duan et al., 2013*). Immunocytochemical analysis of FACS-sorted BFP$^+$ cells confirmed highly enriched expression of LMX1A (95.51±0.09%) and FOXA2 (95.35±0.14%), and co-expression of both markers (94.68±0.10%; *Figure 1B and D*). In contrast, only a small number of LMX1A$^+$ cells (4.96±0.70%) were present in the sorted BFP$^-$ population (*Figure 1B and D*). These findings provide further support that BFP expression faithfully recapitulates LMX1A expression, and that LMX1A$^+$ ventral midbrain progenitors represent the major cell population at d19 (*Cardo et al., 2023*).

The d19 cells were then induced to undergo astrogenic switch in media containing FGF2 and EGF with the BFP$^+$ cells (LMX1A$^+$ ventral midbrain progenitors or their derivatives) monitored by flow cytometry at each weekly passaging (*Figure 1A*, representative gating strategy is shown in *Figure 1—figure supplement 1B–E*). Unexpectedly, we found a dramatic decrease in the BFP$^+$ cell proportion from the starting point of 88.20±1.10% to only 27.59±8.28% at d43 and nearly absent as differentiation continued (*Figure 1E*). We did not observe any evident cell death during culture and replating; thus, the absence of BFP could be either due to the silencing of BFP expression in the derivatives of LMX1A$^+$ progenitors or the loss of these cells through growth competition. To address this question, we performed progenitor expansion and astrogenic induction under the same culture conditions as purified d19 BFP$^+$ progenitors isolated using fluorescence-activated cell sorting (FACS). Interestingly, the proportion of BFP$^+$ cells remained at approximately 90% throughout the astrogenic induction and glial progenitor expansion periods (*Figure 1E*). This observation demonstrates that BFP expression can be maintained in the derivatives of LMX1A$^+$ midbrain progenitors and that the loss of BFP$^+$ cells in the unsorted culture is likely due to their growth disadvantage compared to the derivatives of LMX1A$^-$ progenitors. Our findings are unexpected and demonstrate that the regional or lineage identity of PSC-derived astrocytic cells should not be assumed based merely on the dominant lineage identity of their cellular origin, given that no in vitro fate specification paradigm is 100% efficient.

## Astrogenic switch occurred earlier in derivatives of LMX1A$^+$ midbrain progenitors

Since sorted BFP$^+$ (LMX1A$^+$ ventral midbrain progenitors or their derivatives) and unsorted astrocytic cultures differ distinctively in BFP expression soon after the initiation of astrogenic induction, for simplicity, these cultures are hereafter referred to as the BFP$^+$ and BFP$^-$ cultures, respectively. To determine whether the two cell populations behave differently in the process of astrogenic switch, we examined astrocytic marker expression in these cultures at d45 and d98 using immunocytochemistry. SOX9 and NFIA are transcription factors that are crucial for the initiation of astrogenesis and acquisition of astrogenic competence in the developing central nervous system (*Stolt et al., 2003*; *Deneen et al., 2006*), whereas CD44 has been identified as an astrocyte-restricted precursor (*Liu et al., 2004*). We found that all these markers were more abundantly detected in the BFP$^+$ cultures (NFIA: 65.89±2.81%; SOX9: 57.19±4.25%) than in the FP cultures (NFIA: 4.26±1.28%; SOX9: 8.88±1.82%) at d45 (Two-way ANOVA with post-hoc Tukey test, NFIA: p=2.52×10$^{-5}$, SOX9: p=9.21×10$^{-6}$; *Figure 2A and C* with the original images shown in *Figure 2—figure supplement 1*). Although the number of NFIA$^+$ and SOX9$^+$ cells significantly increased in the BFP$^-$ cultures by d98 (NFIA: 44.07±4.56% on d98, p=3.58×10$^{-4}$; SOX9: 44.28±2.84% on d98, p=1.21×10$^{-4}$), the BFP$^+$ cultures still contained more cells

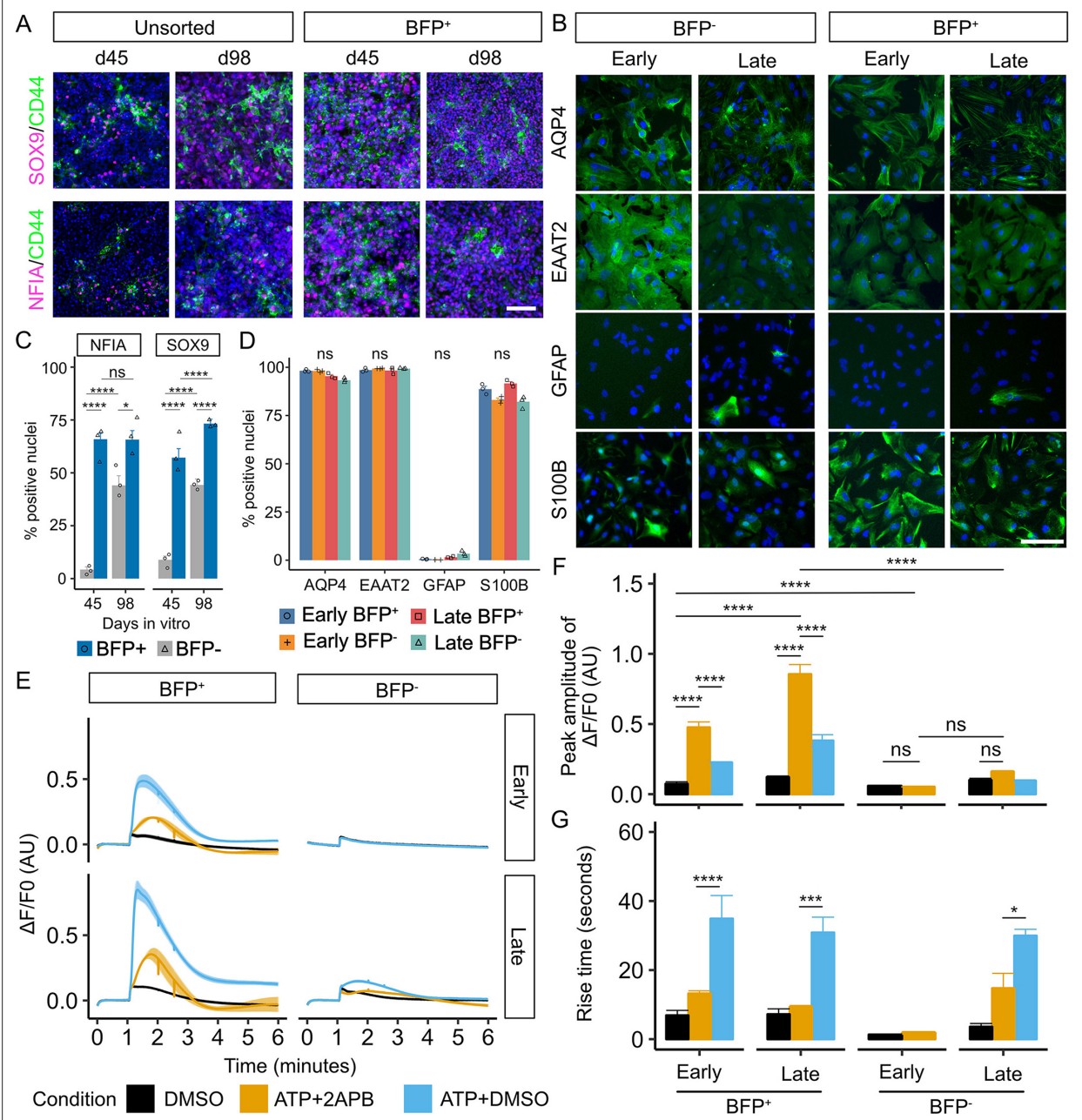

**Figure 2.** Early astrogenic switch and astrocyte maturation in derivatives of LMX1A⁺ midbrain progenitors. (**A**) Representative view of immunocytochemistry of astrogenic marker expression in BFP⁺ and unsorted progenitors at day 45 and 98. Scale bar represents 100 μm. Images shown were cropped to 462 μm×462 μm by randomly selecting the region of interest in the nuclei-only channel (uncropped grayscale images are shown in *Figure 2—figure supplement 1*). (**B**) Representative view of immunocytochemistry of astrocyte marker expression in early and late, BFP⁺ and BFP⁻ (unsorted) astrocytes. Scale bar represents 100 μm. Images shown were cropped to 300 μm×300 μm by randomly selecting the region of interest in the nuclei-only channel (uncropped greyscale images are shown in *Figure 2—figure supplement 2*). (**C**) Quantification of immunocytochemistry of astrogenic marker in BFP⁺ and BFP⁻ progenitors at day 45 and 98 shown in Panel A. Error bars represent the standard error of means (SEM) of three independent experiments. Two-way ANOVA was performed to compare between lineages (NFIA: p=5.389×10⁻⁶, df = 1, effect size = 3.62; SOX9: p=1.96×10⁻⁶, df = 1, effect size = 4.77) and days of differentiation (NFIA: p=7.82×10⁻⁵, df = 1, effect size = 1.99; SOX9: p=2.62×10⁻⁵, df = 1, effect size = 2.99) (**D**) Quantification of immunocytochemistry of astrocyte marker expression in astrocytes. Error bars represent SEM of three independent experiments. Kruskal-Wallis test results following Bonferroni correction are shown on the top of the figure (AQP4: p.adjust=0.12, df = 3, H=8.95; EAAT2: p.adjust=1.00, df = 3, H=0.95; GFAP: p.adjust=0.06, df = 3, H=10.38; S100B: p.adjust=0.11, df = 3, H=9.05). (**E**) Averaged trace of ATP-induced Ca²⁺ response assayed using FLIPR. Drugs or DMSO were applied at 1 min of the assay. The line represents the average fluorescence change (ΔF/F0) in at least three independent experiments, each with at least three replicate wells. The shaded area represents the SEM across at least three independent

*Figure 2 continued on next page*

*Figure 2 continued*

experiments. (**F**) Quantitative comparison of the peak amplitude of ATP-induced $Ca^{2+}$ response among conditions (two-way ANOVA, $p<2.2\times10^{-16}$, df = 2, effect size = 2.54) and samples ($p=2.87\times10^{-14}$, df = 3, effect size = 2.17). Error bars represent the SEM across at least independent experiments. (**G**) Quantitative comparison of the rise time of ATP-induced $Ca^{2+}$ response among conditions (two-way ANOVA, $p=2.19\times10^{-13}$, df = 2, effect size = 1.958) and samples ($p=0.064$, df = 3, effect size = 0.76). Intergroup comparison was performed using post-hoc Tukey test. Error bars represent the SEM across at least three independent experiments. (****$p<0.0001$, ***$p<0.001$, **$p<0.01$, *$p<0.05$, ns: not significant).

The online version of this article includes the following figure supplement(s) for figure 2:

**Figure supplement 1.** Original images of immunocytochemistry of astrogenic markers shown in *Figure 2A*.

**Figure supplement 2.** Original images of immunocytochemistry of astrocyte markers shown in *Figure 2B*.

**Figure supplement 3.** Non-patterned astrocytes.

expressing NFIA ($65.71\pm4.25\%$; $p=2.06\times10^{-2}$) and SOX9 ($73.25\pm2.12\%$; $p=5.51\times10^{-4}$) than in BFP⁻ cultures (*Figure 2A and C*).

To investigate whether the temporal difference in the astrocytic switch between the BFP⁺ and BFP⁻ cultures affects the maturation and functionality of the derived astrocytes, we initiated astrocyte terminal differentiation by exposing the BFP⁺ and BFP⁻ astrocyte precursors to CNTF and BMP4 (*Krencik et al., 2011*; *Bradley et al., 2019*) from d87 (referred to as early astrocytes) and d136 (late astrocytes). Both BFP⁺ and BFP⁻ cultures exhibited a similar expression profile of classic astrocyte markers, including AQP4, EAAT2, and S100B, but few GFAP⁺ cells at both time points (*Figure 2B and D* with the original images shown in *Figure 2—figure supplement 2*). As a reference, we also generated neural progenitors without employing any patterning cues and induced astrogenic switch and astrocyte differentiation from these non-patterned neural progenitors (*Figure 2—figure supplement 3*). We found that, while the astrocyte cultures derived from the non-patterned progenitors contained a similar proportion of cells expressing AQP4, EAAT2, and S100B compared to the BFP⁺ and BFP⁻ astrocyte cultures, there are more GFAP⁺ cells in the non-patterned astrocyte preparations ($17.89\pm5.4\%$, *Figure 2—figure supplement 3*).

Functional astrocytes exhibit transient calcium ($Ca^{2+}$) spikes upon chemical stimulation such as ATP (*Zhang et al., 2016*). Using a FLIPR $Ca^{2+}$ release assay, we observed a sharp increase in the intracellular $Ca^{2+}$ concentration upon ATP administration in both the early and late BFP⁺ astrocyte populations (early BFP⁺: $p=5.37\times10^{-12}$; late BFP⁺: $p<2.2\times10^{-16}$; *Figure 2E and F*). ATP-induced $Ca2^+$ release is partially mediated by inositol trisphosphate. Indeed, addition of an inositol trisphosphate receptor antagonist 2-aminoethoxydiphenylborate (2-APB) reduced the amplitude (early BFP⁺: $p=5.88\times10^{-5}$; late BFP⁺: $p=9.22\times10^{-10}$; *Figure 2F*) and rise time (early BFP⁺: $p=2.63\times10^{-5}$; late BFP⁺: $p=3.59\times10^{-4}$; late BFP⁻: $p=0.028$; *Figure 2G*) of ATP-induced $Ca2^+$ response in both the early and late BFP⁺ astrocytes. Interestingly, early BFP⁺ astrocytes had a significantly lower peak amplitude than that observed in late BFP⁺ astrocytes ($p=6.92\times10^{-9}$; *Figure 2F*), despite their similar levels of astrocyte marker expression, suggesting a difference in maturity at the functional level. Early and late BFP⁻ astrocytes exhibited a similar profile of time-dependent increase in the amplitude of ATP-induced $Ca^{2+}$ response but did not reach statistical significance (*Figure 2E and F*). However, late BFP⁻ astrocytes showed a significantly lower peak amplitude than late BFP⁺ astrocytes ($p<2.2\times10^{-16}$; *Figure 2E and F*).

Taken together, our data demonstrated that astrogenesis occurred earlier in BFP⁺ cultures than in BFP⁻ cells. This temporal difference is also reflected in the functional maturity of the derived astrocytes despite a similar expression profile of classic astroglial markers.

## Single-cell RNA sequencing confirms the authenticity of PSC-derived astrocytes

To further characterize the PSC-derived astrocytes, we performed full-length scRNAseq on early and late BFP⁺ and BFP⁻ astrocytes using the iCELL8 platform and SMART-seq technology, with non-patterned astrocytes derived from the LMX1A-Cre/AAVS1-BFP tracer line as a control. A sample of iPSC-derived neurons was included to facilitate the downstream cell type identification. We profiled FACS-purified astrocytes expressing CD49f as well as unsorted cultures for comparison of all three astrocyte populations (*Figure 3—figure supplement 1*; *Barbar et al., 2020*). After stringent filtering (*Figure 3—figure supplement 2*, see Methods and materials on filtering), we obtained 17478

protein-coding genes in 1786 qualifying cells, with an average of 6326 protein-coding genes detected per cell.

Unsupervised Louvain clustering identified 12 cell clusters (*Figure 3A*). Cells were clustered mainly based on sample type (astrocytes and neurons; *Figure 3—figure supplement 3*) and the estimated cell cycle phase (*Figure 3—figure supplement 3*), whereas sorted CD49f[+] and unsorted astrocytes were largely intermingled together (*Figure 3—figure supplement 3*). Clusters 0, 1, 4, 5, and 11 consisted of predominantly BFP[+] astrocytes, each with varying contributions from the 'early' or 'late' astrocyte samples. Clusters 2, 6, and 7 were dominated by BFP-astrocytes, with the majority of the 'early' astrocyte samples in cluster 2. Cells in clusters 8, 9, and 10 were primarily non-patterned astrocytes, whereas cluster 3 came from the neuronal sample (*Figure 3B and C*). Consistent with immunocytochemistry, *TagBFP* transcripts were detected at a higher level in BFP[+] astrocyte samples than in BFP[-] and NP samples, while *TagBFP* expression was negligible in neuronal samples derived from an iPSC line without the BFP transgene (*Figure 3—figure supplement 3*). Using a set of known astrocyte and neuronal signature genes (*Figure 3D*), we identified cells in clusters 0, 1, and 5–11 as astrocytes (*Figure 3D*), which were enriched in SOX9, NFIA, NFIB, HES1, S100A13, S100A16, EGFR, CD44, and GJA1 expression (*Figure 3D*). These transcripts were also detected at high levels in clusters 2 and 4, which were mostly estimated to be in the cell cycle phases G2, M, and S (*Figure 3—figure supplement 3*). In addition, clusters 2 and 4 showed high levels of proliferation-related transcripts, such as *TOP2A, MKI67, CDK1*, and *AURKA* (*Figure 3D*) and were thus defined as astrocyte precursors. In contrast, Cluster 3 contained mostly cells from the neuronal sample (*Figure 3A, Figure 3—figure supplement 3*) and expressed high levels of genes closely related to neuronal structure and function (such as *STMN2, SYT1, DCX, MAPT*, and *SNAP25*; *Figure 3D*). We did not detect transcripts indicative of endoderm (*GATA4*), mesoderm (*TBXT* and *TBX6*), or oligodendrocyte progenitors (*SOX10* and *PDGFRA*) in any of these clusters (*Figure 3—figure supplement 3*).

Next, we examined the developmental status of astrocytes by pseudobulk analysis using published bulk RNA-seq datasets from human fetal and postmortem astrocytes (*Zhang et al., 2016*). As expected, the astrocyte precursors (clusters 2 and 4) were projected to be less mature than the fetal astrocytes (*Figure 3E*). The BFP[+] astrocyte dominant clusters (0, 1, 5, and 11) and non-patterned astrocyte clusters (8, 9, and 10) were shown to be more developmentally advanced than the BFP[-] astrocyte clusters (6 and 7). Thus, the pseudobulk analysis provides independent support to the functional assays, suggesting differences in the relative maturity of BFP[+] and BFP[-] astrocytes (*Figure 2*).

To determine the authenticity of these PSC-derived astrocytes, we mapped our data to five published scRNAseq datasets obtained from human fetal and adult brains using Seurat integration (*La Manno et al., 2016; Sloan et al., 2017; Zhong et al., 2018; Polioudakis et al., 2019; Agarwal et al., 2020; Fan et al., 2020; Bhaduri et al., 2021; Eze et al., 2021*). We found that cells in clusters annotated as astrocytes (clusters 0, 1, and 5–11) were predominantly mapped to the reference astrocyte or astrocyte precursor populations with high confidence (prediction score >0.5; *Figure 3F, Figure 3—figure supplement 4*). In contrast, neuronal cluster 3 was mapped to neurons in the fetal reference datasets, while the astrocyte precursor clusters (2 and 4) were mapped to progenitor populations in the fetal reference datasets (*Figure 3F, Figure 3—figure supplement 4*). These findings demonstrate that iPSC-derived astrocytes closely resemble those in the human brain.

## Distinct transcriptome fingerprints of LMX1A[+] midbrain progenitor-derived astrocytes

Significant advances have been made recently in the understanding of the molecular profiles of midbrain dopamine neurons. However, our knowledge of midbrain astrocytes in this regard remains limited and does not inform the anatomic or lineage origin of the cells. In this regard, BFP[+] astrocytes provide a unique resource for determining the transcriptomic characteristics of human astrocytes derived from the LMX1A[+] lineage of ventral midbrain patterned progenitors. By performing pairwise differential gene expression (details described in Methods and materials), we identified 1153 genes differentially expressed (DEGs; adjusted p values less than 0.05 and log2 fold change over 0.25) in BFP[+] astrocytes when compared to either BFP[-] or non-patterned astrocyte populations (*Supplementary file 1A*). Of these, 287 were unique to BFP[+] astrocytes (BFP[+] enriched, *Figure 4A*), including genes associated with midbrain dopamine neuron development such as *SULF1, LMO3, NELL2*, and *RCAN2* (*Figure 4B; Strelau et al., 2000; La Manno et al., 2016; Bifsha et al., 2017; Ahmed et al.,*

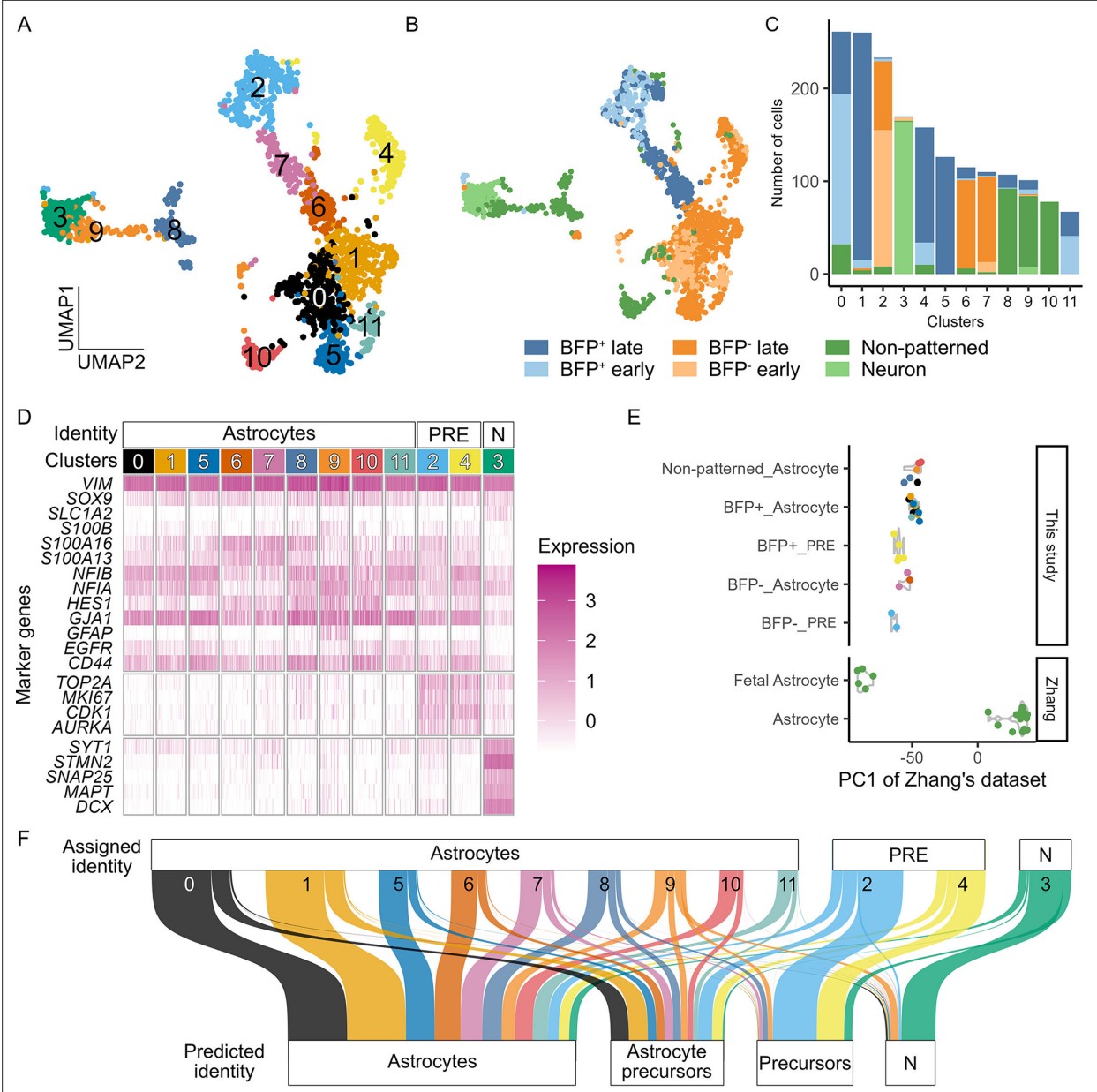

**Figure 3.** Single-cell RNA sequencing confirms the authenticity of pluripotent stem cell (PSC)-derived astrocytes. (**A, B**) Uniform manifold approximation and projection plot of unbiased clustering, coloured by clusters or sample group. (**C**) Number of cells from different sample groups in each clusters. (**D**) Heatmap of the normalized expression of selected markers in different clusters. The assigned identity to each cluster is shown at the top of the plot. (**E**) Principal component projection analysis of pseudobulk astrocyte data onto a reference principal component axis of astrocyte maturity (**Zhang et al., 2016**; **Hedegaard et al., 2020**). Each dot represents a pseudobulk sample of one independent sample from each clusters. (**F**) Sankey plot summarizing the result of reference mapping of cells in different clusters to eight published reference human brain scRNAseq datasets. The thickness of the thread is proportional to the number of cells mapped to the same identity in the reference datasets (predicted identity). Detailed results of referencing mapping to each reference datasets are shown in **Figure 3—figure supplement 4A-H** and prediction score shown in **Figure 3—figure supplement 4I**. (PRE: precursors; N: neurons).

The online version of this article includes the following figure supplement(s) for figure 3:

**Figure supplement 1.** Fluorescence-activated cell sorting of astrocytes for single-cell RNA sequencing.

**Figure supplement 2.** Processing of single-cell RNA sequencing data.

**Figure supplement 3.** Expression of marker genes as detected by scRNAseq.

**Figure supplement 4.** Reference mapping to human brain single-cell RNA sequencing datasets.

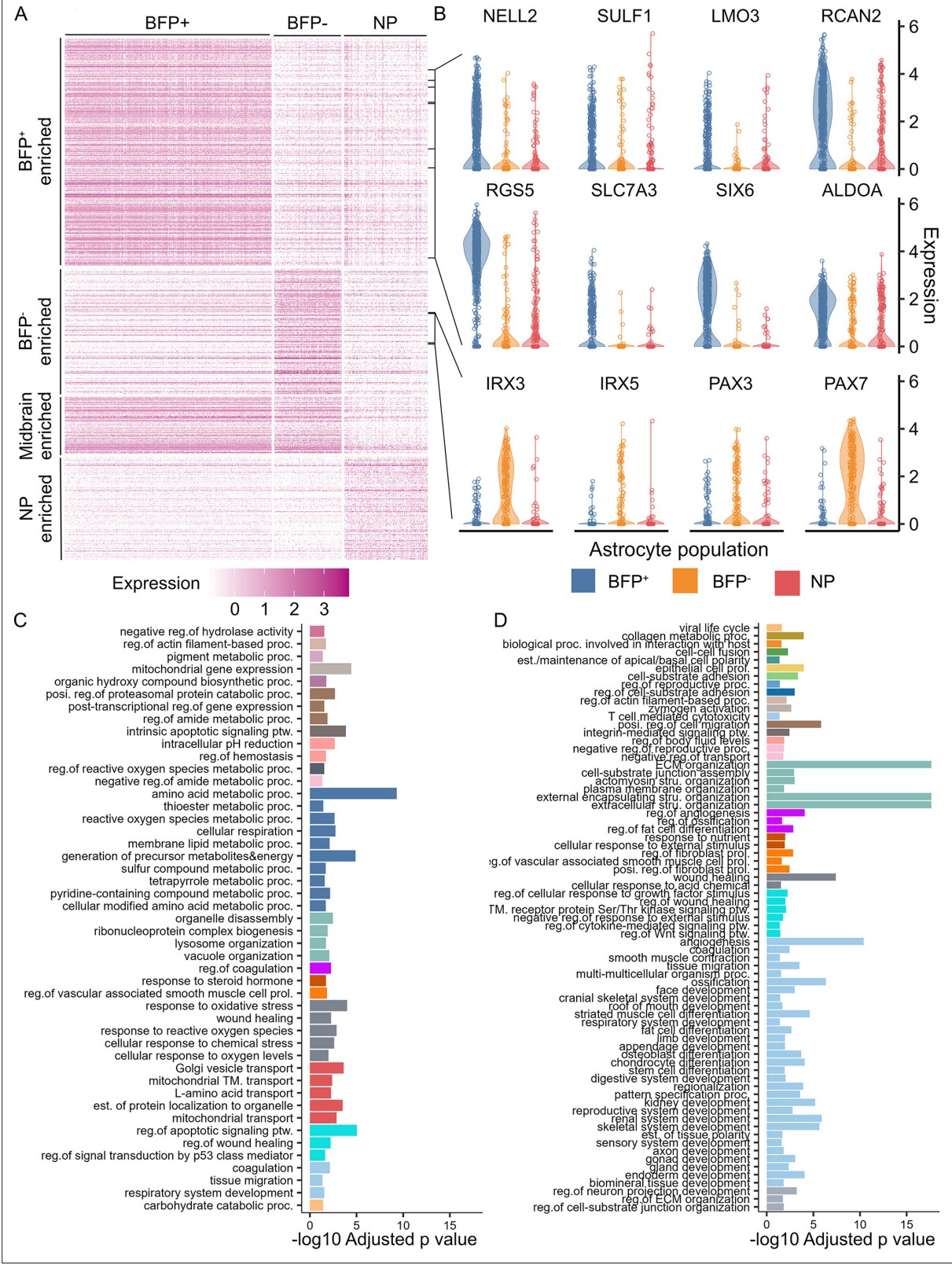

**Figure 4.** Distinct transcriptome fingerprints of astrocytes derived from LMX1A+ ventral midbrain progenitors. (**A**) Heatmap of the normalized expression of population-specific genes in different populations of astrocytes. (**B**) Violin plots of the normalized expression of selected candidate markers for BFP+, BFP-, and non-patterned (NP) astrocytes. (**C, D**) Representative GO terms significantly enriched in BFP+ (**C**) and BFP- (**D**) enriched genes. Semantically similar representative terms were shown with the same color.

*Figure 4 continued on next page*

*Figure 4 continued*

The online version of this article includes the following figure supplement(s) for figure 4:

**Figure supplement 1.** Validation of candidate markers for BFP⁺ and BFP⁻ astrocytes.

*2021*). 55 of the 287 BFP⁺ DEGs, including *SULF1*, *NELL2*, *RCAN2*, and *RGS5,* were confirmed to be significantly preferentially expressed in midbrain astrocytes in vivo (adjusted p-values less than 0.05) by analysing the integration of five human brain datasets (*Figure 4—figure supplement 1*; *La Manno et al., 2016*; *Zhong et al., 2018*; *Fan et al., 2020*; *Eze et al., 2021*). Another 87 BFP⁺ DEGs, such as *SLC7A3, SIX6,* and *LMO3*, were also detected at higher levels in midbrain astrocytes than in forebrain astrocytes, although their expression levels were not significant (*Figure 4—figure supplement 1*). Moreover, *LMX1A* and *FOXA2*, which have been used to evaluate PSC-derived midbrain astrocytes in previous studies, were not detected in BFP⁺ astrocytes (*Figure 3—figure supplement 3*, *Figure 4— figure supplement 1*).

We also identified 159 DEGs enriched only in BFP⁻ astrocytes (BFP⁻ enriched, *Figure 4A* and *Supplementary file 1A*), including those known to be expressed in the ventrolateral-dorsal domain of the midbrain and hindbrain, such as *IRX3*, *IRX5*, *PAX3*, and *PAX7* (*Figure 4B*; *Houweling et al., 2001*; *Matsunaga et al., 2001*). Differential expression of PAX3 and PAX7 in BFP astrocytes was confirmed at the protein level by immunocytochemistry of d136 BFP⁺ and BFP⁻ astrocyte cultures, whereas their expression in the human midbrain was validated by published datasets in silico (*Figure 4—figure supplement 1*). This transcription profile supports the notion that BFP astrocytes are descendants of initial minor populations of lateral midbrain progenitors. Moreover, 72 DEGs were shared by BFP⁺ and BFP⁻ astrocytes compared to non-patterned astrocytes (midbrain enriched, *Figure 4A* and *Supplementary file 1A*). This set of genes included *NR2F1*, *NR2F2*, *ZEB2*, *KCNJ6*, and *SRPX* (*Figure 4B*), which have been reported to be signatures of mouse midbrain astrocytes (*Endo et al., 2022*). Together, our findings provide a new entry into the transcriptomic characteristics of midbrain astrocytes, specifically a gene expression map of the LMX1A⁺ midbrain progenitor-derived human astrocyte lineage.

Gene ontology (GO) enrichment analysis was performed on the 1153 DEGs enriched in BFP⁺ astrocytes (*Supplementary file 1B*). The significantly enriched GO terms were mainly related to various aspects of metabolism, stress response, biosynthesis, lysosomal activity, and cellular respiration (*Figure 4C* and *Supplementary file 1C*). These biological processes have previously been shown to be disrupted by several mutations that cause familial Parkinson's disease (*di Domenico et al., 2019*; *Barbuti et al., 2020*; *Sonninen et al., 2020*). In contrast, the GO terms associated with 530 BFP⁻ astrocyte-enriched DEGs were mostly related to the formation of the extracellular matrix and tissue development (*Figure 4D* and *Supplementary file 1B and C*). The differential enrichment of GO terms implies functional differences between BFP⁺ and BFP⁻ astrocytes and supports the need for generating region-specific astrocytes for disease modelling.

## Discussion

Despite the general belief that recapitulating astrocyte lineage heterogeneity is necessary for stem cell-based disease modelling and cell transplantation, the extent of astrocyte heterogeneity in different brain regions, their anatomical origins, and associated molecular signatures remain largely elusive. This knowledge gap limits the endpoint characterization of stem cell-derived astrocytes; hence, reliance on the dominant regional characteristics of the initial neural progenitor populations, with the assumption that the lineage representation of the progenitors is preserved in the derived astrocytes. By harnessing an LMX1A-based lineage tracing human iPSC line, we discovered unexpected negative selection against derivatives of LMX1A⁺ midbrain-patterned progenitors during astrocyte induction and progenitor expansion, highlighting the need for careful characterization of PSC-derived astrocytes and reinforce the need for a deeper understanding of the molecular landscape of astrocytes in different regions of the human brain.

Most neural progenitors used for astrocyte differentiation in this study coexpressed LMX1A and FOXA2. In the ventral midbrain, the combinatorial expression of these transcription factors defines the dopaminergic neural progenitors (*Failli et al., 2002*; *Andersson et al., 2006*). We found that astrocytes derived from LMX1A⁺ progenitors could only be obtained if LMX1A⁺ cells were purified prior to astrocyte differentiation. In contrast, astrocytes derived from bulk midbrain-patterned

progenitors exhibit transcriptomic profiles of the lateral-dorsal midbrain despite LMX1A⁺ progenitors being the predominant starting population. Our findings demonstrate that the lineage composition of parent progenitors was not preserved during astrocyte induction or progenitor expansion. FGF is the most used inductive molecule for astrocyte differentiation from stem cells (*Chandrasekaran et al., 2016*). However, it is evident that FGF-expanded neural progenitors, originating either from the brain or neutralized PSCs, exhibit restricted regional competence and positional gene expression. For example, bulk-expanded human ventral midbrain neural progenitors (*Jain et al., 2003*), fetal forebrain or spinal cord-derived neural stem (NS) cells only give rise to GABAergic neurons (*Sun et al., 2008*), and lt-NES cells display an anterior hindbrain-like positional profile (*Falk et al., 2012*), while their antecedents, PSC-derived neural rosettes and early passage derivatives, express anterior forebrain markers (*Koch et al., 2009*). It is unclear whether this is due to the deregulation of the original patterning at the level of gene expression or the loss of the associated cell population (*Gabay et al., 2003*). In this study, because BFP⁺ astrocytes can be generated under the same culture conditions as purified LMX1A⁺ progenitors, we reasoned that the loss of their derivatives in unsorted cultures was possibly due to their differential growth capacity.

Our study highlights the need for a careful assessment of the positional identity of in vitro-derived astrocytes. A common practice in this regard is to confirm the regional identity of founder progenitors following fate-directed neural induction, with the assumption that the dominant positional features are maintained by astrocyte progeny (*Krencik et al., 2011*). This strategy is, at least partly, dictated by our limited knowledge of the gene expression signatures of regional and/or lineage-specific astrocytes. Hence, endpoint evaluation of PSC-derived astrocytes often relies on region-specific markers defined in the brain during the neurogenic period. For example, LMX1A and FOXA2 expression has been used as criteria for midbrain astrocytes in previous studies (*Barbuti et al., 2020*; *Crompton et al., 2023*). However, scRNA-seq of the human fetal ventral midbrain and adult substantia nigra has revealed negligible expression of these transcripts in astrocytes (*La Manno et al., 2016*; *Agarwal et al., 2020*; *Kamath et al., 2022*). Consistent with these findings, we did not detect LMX1A or FOXA2 in BFP⁺ or BFP⁻ astrocytes. However, our analysis identified new positive and negative markers that can be used to validate astrocytes derived from the LMX1A⁺ lineage of ventral midbrain progenitors. Future work will benefit from in vivo validation of these putative lineage-specific markers, as a mouse analogous tracer line is available, whereas lineage tracking is not possible in humans.

In addition to the distinct transcriptomic profiles, BFP⁺ and BFP astrocytes may also be functionally different. Astrocytes generated from progenitors broadly patterned to the dorsal forebrain, ventral forebrain, and spinal cord have been shown to exhibit different GO enrichment profiles as well as different physiological and functional properties (*Bradley et al., 2019*). In comparison to BFP⁻ and non-patterned astrocytes, the current study revealed that GO terms enriched in BFP⁺ astrocytes, which originated from the same progenitor giving rise to midbrain dopaminergic neurons, were closely related to various biological processes disrupted in astrocytes carrying familial Parkinson's disease mutations (*di Domenico et al., 2019*; *Barbuti et al., 2020*; *Sonninen et al., 2020*). Such a distinct enrichment profile suggests that BFP⁺ astrocytes may be functionally adapted to support midbrain dopaminergic neurons compared with BFP⁻ and non-patterned astrocytes. Indeed, astrocytes isolated from the ventral midbrain have been reported to exhibit stronger neurotrophic effects and the ability to reduce α-synuclein aggregation in the midbrain than cortical astrocytes in cellular and mouse models of PD (*Kostuk et al., 2019*; *Yang et al., 2022*), highlighting the importance of understanding astrocyte heterogeneity in iPSC disease modelling.

In conclusion, this study provides further evidence for the regional diversity of astrocytes and identifies a set of midbrain-enriched genes. Crucially, the transcriptomic fingerprint of human astrocytes derived from LMX1A-expressing midbrain progenitors reported here offers a much-needed resource for assessing the authenticity of stem cell-derived astrocytes in studies of Parkinson's disease.

## Materials and methods
### Stem cell culture and astrocyte differentiation
KOLF2 human iPSCs (HPSI0114i-kolf2; https://hpscreg.eu/cell-line/WTSIi018-B; *Hildebrandt et al., 2019*) were obtained from the Sanger Institute. The cell line identity was confirmed by in-house SNP analysis using the Infinium Psych Array v1.1. Cells were routinely tested to ensure they were free

from mycoplasma contamination and maintained chromosomal integrity. Undifferentiated iPSCs were maintained in E8 flex media (Thermo Fisher) and manually dissociated using Gentle Cell Dissociation Reagent (STEMCELL Technologies) as previously described (*Cardo et al., 2023*). Astrocytes were differentiated using a three-stage stepwise strategy consisting of neural induction and regional patterning, astrogenic switch and progenitor expansion, and astrocyte terminal differentiation. LMX1A⁺ ventral midbrain progenitors were generated as previously described (*Nolbrant et al., 2017*; *Cardo et al., 2023*). At day 19, cells were replated as single cells onto poly-D-lysine-laminin-coated plates at 1×10⁶ cells/cm² for astrogenic switch and progenitor expansion in N2B27 media supplemented with 10 ng/mL FGF2 (Peprotech) and 10 ng/mL Human EGF (Peprotech) and replated every 6–8 days. For astrocyte terminal differentiation, expanded neural progenitors were re-plated at a density of 3×10⁴ cells/cm² in expansion media and 24 hours later switched to N2B27 supplemented with 10 ng/mL human recombinant CNTF (Peprotech) and 10 ng/mL human recombinant BMP4 (Peprotech) for 7 days followed by media containing CNTF alone for another 13 days. 10 µM Y-27632 was used for 24 hr before and after each replating. The protocol for generating non-patterned astrocytes was the same as for LMX1A⁺ ventral midbrain-derived astrocyte, except the neural progenitors were derived with duo-SMAD inhibitors only without ventral patterning reagents.

## Flow cytometry analysis and cell isolation

Cells were dissociated in Accutase as described above and washed twice with DPBS by centrifugation for 5 min at 200 rcf. For evaluating BFP expression, dissociated cells were resuspended in 0.5 mM EDTA in DPBS (Sigma-Aldrich) and analyzed on a BD LSRFortessa cell analyser (BD Biosciences). For purifying BFP⁺ cells, dissociated cells were resuspended in the same cell culture media. Background autofluorescence was compensated for using KOLF2 parental cell line at a similar stage of differentiation to define BFP⁻ gating. For purifying CD49f⁺ astrocytes, dissociated cells were stained with Alexa Fluor 647-conjugated rat anti-CD49f antibody (5% v/v in a 100 µL reaction; BD Biosciences) for 25 min at 37 °C on an orbital shaker at 200 rcf and resuspended in DPBS containing 0.5% bovine serum albumin and 50 units/mL DNase I (Sigma-Aldrich). Background autofluorescence was compensated for using KOLF2 parental cell line at a similar stage of differentiation to define BFP⁻ gating and unstained astrocytes to define CD49f⁻ gating. Cell sorting was performed on a BD FACSAria III (BD Biosciences) using an 80 µm nozzle. Sorted cells were collected in the same media as for resuspension. Flow cytometry data were analyzed in FlowJo v10.8.1 (BD Biosciences) as shown in *Figure 1—figure supplement 1B–E*. Briefly, non-debris events were selected using the eclipse gates on dot graphs of SSC-A versus FSC-A. Singlet events were sequentially gated using polygonal gates on dot graphs of FSC-H versus FSC-A and SSC-H versus SSC-A by selecting the events in the diagonal region. The positive and negative gates in the fluorescence channel were set as bifurcate gates at a minimum of 99.9% percentile (usually at 99.99% percentile) on the histogram of the fluorescence intensity of the negative control sample of the same flow cytometry experiment and applied to all samples of the same flow cytometry experiment.

## FLIPR calcium assay

Astrocytes were plated on day 10 of astrocyte terminal differentiation and cultured according to the standard protocol until day 20. On the day of recording, an equal volume of FLIPR Calcium 6 Assay buffer was added to cells without replacing or washing and cells were incubated for 2 hr at 37 °C. Drug assay buffers were prepared in HBSS with $Ca^{2+}$ and $Mg^{2+}$ (Gibco) with 20 mM HEPES buffer (Gibco) at 5 X concentration. After incubation, cells were imaged on FLIPR Penta system (Molecular Devices) at a frequency of 2 Hz for 1 min prior to injection. 25 µL of the drug assay buffers were dispensed at a speed of 25 µL/s and images captured at a frequency of 2 Hz for a further 5 min. Raw fluorescence intensity data were exported and analyzed in R 4.3.0 using a custom script to identify the peak response. The average baseline fluorescence intensity for each well was calculated by averaging the raw fluorescence intensity measured 60 s prior to drug application. The normalized change in fluorescence intensity above the baseline (ΔF/F0) for each well, at a given timepoint, was calculated using the following formula:

$$\Delta\text{F/F0} = \frac{F_t - F_0}{F_0}$$

where $F_t$ is the fluorescence intensity at timepoint t, $F_0$ is the average fluorescence intensity of the baseline period (60 s prior to drug application).

## Immunocytochemistry

Cultures were fixed with 3.7% PFA for 15–20 min at 4 °C. For nuclear antigen detection, an additional fixation with methanol gradient was performed, which include 5 min each in 33% and 66% methanol at room temperature followed by 100% methanol for 20 min at –20 °C. Cultures were then returned to PBST via inverse gradient and were then permeabilized with three 10 min washes in 0.3% Triton-X-100 in PBS (PBS-T) and then blocked in PBS-T containing 1% BSA and 3% donkey serum. Cells were incubated with primary antibodies in blocking solution overnight at 4 °C. Following three PBS-T washes, Alexa-Fluor secondary antibodies (Thermo Fisher Scientific) were added at 1:1000 PBS-T for 1 hr at ambient temperature in the dark. Three PBS-T washes were then performed that included once with DAPI (Molecular Probes). Images were taken on a Leica DMI6000B inverted microscope. Quantification was carried out in Cell Profiler (*Stirling et al., 2021*) or manually using ImageJ (*Schindelin et al., 2012*) by examining at least four randomly selected fields from three independent experiments. The antibodies used are provided in the *Supplementary file 2*. Representative images shown in main figures were cropped by randomly selecting the region of interest in the DAPI-stained channel only, with the original unedited images shown in *Figure 1—figure supplement 1*, *Figure 2—figure supplements 2 and 3*.

## Single-cell RNA sequencing

Cells were dissociated with Accutase with 10 units/mL of papain (Sigma-Aldrich) for 10 min at 37 °C and resuspended in 0.5% bovine serum albumin (Sigma-Aldrich) with 50 units/mL DNase I in DPBS without calcium or magnesium (Gibco) and passed through a cell strainer with 35 μm mesh. Cells were stained with 1 μM SYTO16 (Invitrogen) and 0.08% (v/v) propidium iodide (part of the Invitrogen ReadyProbes Cell Viability Imaging Kit, Blue/Red) for 20 min on ice and then dispensed into the nano-well plates of the ICELL8 cx Single-Cell System (Takara). Wells containing single viable cells were automatically selected using the ICELL8 cx CellSelect v2.5 Software (Takara) with the green and not red logic. Manual triage was performed to recover additional candidate wells that contain viable single cells. The library for sequencing was then prepared using the SMART-Seq ICELL8 application kit (Takara) following the manufacturer's recommended protocol. Next-generation sequencing was performed using the NovaSeq 6000 and the Xp Workflow on a S4 flow cell for 200 cycles of pair-end sequencing.

## Single-cell RNA-sequencing analysis

Using the Cogent NGS Analysis Pipeline software v 1.5.1 (Takara), FASTQ files containing all indices for each chip were demultiplexed to FASTQ files containing one index per file. Adaptor sequences were removed using cutadapt 3.2 with the following settings: `-m 15 --trim-n --max-n` 0.7 -q 20. Trimmed FASTQ files were aligned to the *Homo sapiens* GRCh38.106 primary assembly with the BFP reporter gene attached to the end of the genome, using STAR 2.7.9 a (*Dobin et al., 2013*) with the following settings: `--outSAMtype BAM Unsorted --quantMode TranscriptomeSAM --outReadsUnmapped Fastx --outSAMstrandField intronMotif --chimSegmentMin 12 --chimJunctionOverhangMin 8 --chimOutJunctionFormat 1 --alignSJDBoverhangMin 10 --alignMatesGapMax 100000 --alignIntronMax 100000 --alignSJstitchMismatchNmax 5-1 5 5 --chimMultimapScoreRange 3 --chimScoreJunctionNonGTAG -4 --chimMultimapNmax 20 --chimNonchimScoreDropMin 10 --peOverlapNbasesMin 12 --peOverlapMMp 0.1 –alignInsertionFlush Right --alignSplicedMateMapLminOverLmate 0 --alignSplicedMateMapLmin 30`. After alignment, gene-level quantification was performed using featureCounts from subread 2.0.0 (*Liao et al., 2013*) with the following settings: -t exon `--primary` -R CORE -F GTF -Q 0 -B -g gene_id. The count matrix of each index was combined in R 4.2.0 (*R Development Core Team, 2023*).

All downstream analysis was performed in R 4.3.0 using Seurat 4.3.0 (*Stuart et al., 2019*). Gene-level filtering was applied by including only protein-coding genes with at least five total counts across all cells and being expressed in at least 1% of all cells. Poor quality cells were then identified using the *is.outlier* function from the scater 1.28.0 (*McCarthy et al., 2017*). Poor quality cells were

defined as having a high percentage of mitochondrial gene count, or high or low the total number of genes detected, or high total gene counts. The thresholds of each metrics for each sample were determined as twice the median absolute deviation of the sample. Raw gene counts were log normalized with a scale.factor setting of $1\times10^5$. Data from the two batches of experiments were integrated using the *FindIntegrationAnchors* and *IntegrateData* based on the common top 2000 highly variable genes and the first 30 dimensions of principal components. The percentage of mitochondrial gene count and total gene count were regressed out using the *ScaleData* function. Principal component analysis (PCA) was performed on the top 2000 high variable genes and the number of principal components used for uniform manifold approximation and projection (UMAP) was determined using the JackStraw method (*Chung and Storey, 2015*). UMAP and unbiased Louvain clustering was performed on the first 33 principal components. Pairwise differential gene expression analysis was performed using the MAST method (*Finak et al., 2015*) with 'Chip' as the latent variable. Gene ontology enrichment analysis was performed using *enrichGO* function in the clusterProfiler 4.10.0 package (*Stirling et al., 2021*) with all genes in the filtered dataset as the background. GO term database was downloaded using the org.Hs.eg.db 3.18.0 package (*Carlson, 2019*). Revigo v1.8.1 was used to group the representative GO terms based on semantic similarity using a size setting of 0.5, *Homo sapiens* database, and SimRel method for semantic similarity (*Schlicker et al., 2006*; *Supek et al., 2011*).

Published datasets were downloaded from NCBI's Gene Expression Omnibus (*Clough and Barrett, 2016*) and processed in R 4.2.0. Gene-level filtering was performed by retaining only protein-coding genes with more than five total counts across all cells. Gene counts were normalized using the NormalizeData function (scale factor settings are listed in *Supplementary file 3*). PCA was performed based on the top 2000 highly variable genes to obtain the first 50 PCs. Visual inspection of the elbow plot was used to determine the number of PCs for downstream analysis. Batch effect between subjects was evaluated on the two-dimensional PC2~PC1 plot. Where inter-subject batch effect was observed, Harmony integration was performed based on the PCs selected in the previous step (*Supplementary file 3*). UMAP was performed based on either the PCA or Harmony reduction (using the top 30 dimensions), and Louvain clustering was performed (settings shown in *Supplementary file 3*). Cluster identities were verified against the reported annotation where possible. For datasets without detailed annotation published or astrocyte lineage reported, reannotation was performed based on the expression of known markers (*Figure 3—figure supplement 4*, *Supplementary file 3*). Reference mapping was performed using *FindTransferAnchors* and *TransferData* function in Seurat based on the first 30 dimensions of either the PCA or Harmony loadings of the reference dataset. The analysis code used in this study are deposited at https://github.com/zli-cardiff/LMX1A-astrocyete (copy archived at *Li, 2025*).

## Statistical analysis

All data were collected from at least three independent experiments and presented as mean ± standard error of means unless otherwise specified. Data were tested for normality with the Shapiro-Wilk test and for equal variance with the Levene's test before performing statistical analyses by two-way ANOVA with post-hoc Tukey test for multiple comparisons where relevant. Kruskal-Wallis test with post-hoc Dunn's test for pairwise comparison was used where parametric test was not suitable. Effect size was calculated as Cohen's f for ANOVA or eta squared based on the H-statistic for the Kruskal-Wallis test. All statistical tests were performed in R4.3.0.

## Acknowledgements

We would like to thank Mark Bishop and Joanne Morgan for conducting FACS and next generation sequencing, respectively. We thank Kathryn Peall and Laura Abram for providing iPSC-derived neurons for scRNA-seq. We also thank the support of the Supercomputing Wales project, which is part-funded by the European Regional Development Fund (ERDF) via the Welsh Government. This work was supported by the UK Dementia Research Institute, jointly funded by the UK Medical Research Council, Alzheimer's Society, and Alzheimer's Research UK, to CW (MC_PC_17112) and ZL (DRI-TRA2021-02), and a seed corn fund to ZL from the Neuroscience and Mental Health Innovation Institute, Cardiff University. ZL was funded by a UK Dementia Research Institute PhD studentship.

## Additional information

### Funding

| Funder | Grant reference number | Author |
|---|---|---|
| UK Dementia Research Institute | MC_PC_17112 | Caleb Webber |
| UK Dementia Research Institute | DRI-TRA2021-02 | Zongze Li |
| Cardiff University | | Zongze Li |
| UK Dementia Research Institute | PhD studentship | Zongze Li |

The funders had no role in study design, data collection and interpretation, or the decision to submit the work for publication.

### Author contributions

Zongze Li, Conceptualization, Data curation, Software, Formal analysis, Funding acquisition, Validation, Investigation, Methodology, Writing - original draft, Project administration, Writing – review and editing; Lucia Fernandez Cardo, Resources, Supervision; Michal Rokicki, Data curation; Jimena Monzón-Sandoval, Provided guidance and contributed to general discussions on scRNAseq data analysis; Viola Volpato, Provided guidance and contributed to general discussions on scRNAseq data analysis; Frank Wessely, Provided guidance and contributed to general discussions on scRNAseq data analysis; Caleb Webber, Supervision, Funding acquisition, Writing – review and editing; Meng Li, Conceptualization, Resources, Supervision, Writing - original draft, Writing – review and editing

### Author ORCIDs

Lucia Fernandez Cardo ⓘ https://orcid.org/0000-0002-5407-8175
Meng Li ⓘ https://orcid.org/0000-0002-4803-4643

None https://doi.org/10.7554/eLife.96423.3.sa1
Author response https://doi.org/10.7554/eLife.96423.3.sa2

---

## Additional files

### Supplementary files

Supplementary file 1. Consists of three tables in Excel file. Table A: Pairwise DEGs among the three astrocyte populations and the neuron population. Table B: GO enrichment of population-enriched DEGs. Table C: Representative enriched GO terms of DEGs for BFP$^+$ and BFP$^-$ astrocytes.

Supplementary file 2. Information on antibodies used in this study.

Supplementary file 3. Settings used for processing published transcriptomic datasets.

MDAR checklist

### Data availability

The sequencing data discussed in this publication have been deposited in NCBI's Gene Expression Omnibus (*Clough and Barrett, 2016*) and are accessible through a GEO Series accession number GSE252624. The codes are deposited at https://github.com/zli-cardiff/LMX1A-astrocyete (copy archived at *Li, 2025*).

The following dataset was generated:

| Author(s) | Year | Dataset title | Dataset URL | Database and Identifier |
|---|---|---|---|---|
| Li Z, Cardo LF, Rokicki M, Webber C, Li M | 2024 | Single-cell sequencing of human induced pluripotent stem cell-derived ventral midbrain astrocytes | https://www.ncbi.nlm.nih.gov/geo/query/acc.cgi?acc=GSE252624 | NCBI Gene Expression Omnibus, GSE252624 |

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
