## [Editor Report · eLife Assessment]

The manuscript by Li and coworkers analyzed astrocytic differentiation of midbrain floor plate-patterned neural cells originating from human iPS cells, with a LMX1A reporter. This **valuable** work identifies transcriptomic differences at the single-cell level, between astrocytes generated from LMX1A reporter positive or negative cells, as well as non-patterned astrocytes and neurons. The evidence is **solid**, but the paper can be strengthened by further analyses of the transcriptomic data, and astrocytic morphology; also, searching for some of the differentially expressed genes by immunohistochemistry in different regions of the mammalian brain, or in human specimens, would be very informative.

---

## [Referee Report · None]

Editorial note: To ensure a thorough evaluation of the revised manuscript, we invited a third reviewer to assess whether the authors had sufficiently addressed the concerns raised in the initial round of peer review. This additional reviewer confirmed that the authors responded partially to the original reviewers requests. While he/she also provided a set of new comments, these do not alter the original assessment or editorial decision regarding the manuscript. For transparency and completeness, the additional comments are included below.

**Reviewer #3 (Public Review):**

Summary:

In this manuscript, Li and coworkers present experiments generated with human induced pluripotent stem cells (iPSCs) differentiated to astrocytes through a three-step protocol consisting of neural induction/midbrain patterning, switch to expansion of astrocytic progenitors, and terminal differentiation to astroglial cells. They used lineage tracing with a LMX1A-Cre/AAVS1-BFP iPSCs line, where the initial expression of LMX1A and Cre allows the long-lasting expression of BFP, yielding BFP+ and BFP- populations, that were sorted when in the astrocytic progenitor expansion. BFP+ showed significantly higher number of cells positive to NFIA and SOX9 than BFP- cells, at 45 and 98 DIV. However, no significant differences in other markers such as AQP4, EAAT2, GFAP (which show a proportion of less than 10% in all cases) and S100B were found between BFP-positive or -negative, at these differentiation times. Intriguingly, non-patterned astrocytes produced higher proportions of GFAP positive cells than the midbrain-induced and then sorted populations. BFP+ cells have enhanced calcium responses after ATP addition, compared to BFP- cells. Single-cell RNA-seq of early and late cells from BFP- and BFP+ populations were compared to non-patterned astrocytes and neurons differentiated from iPSCs. Bioinformatic analyses of the transcriptomes resulted in 9 astrocyte clusters, 2 precursor clusters and one neuronal cluster. DEG analysis between BFP+ and BFP- populations showed some genes enriched in each population, which were subject to GO analysis, resulting in biological processes that are different for BFP+ or BFP- cells.

Strengths:

The manuscript tries to tackle an important aspect in Neuroscience, namely the importance of patterning in astrocytes. Regionalization is crucial for neuronal differentiation and the presented experiments constitute a trackable system to analyze both transcriptional identities and functionality on astrocytes.

Weaknesses:

The presented results have several fundamental issues, to be resolved, as listed in the following major points:

(1) It is very intriguing that GFAP is not expressed in late BFP- nor in BFP+ cultures, when authors designated them as mature astrocytes.

(2) In Fig. 2D, authors need to change the designation "% of positive nuclei".

(3) In Fig. 2E, the text describes a decrease caused by 2APB on the rise elicited by ATP, but the graph shows an increase with ATP+2APB. However, in Fig. 2F, the peak amplitude for BFP+ cells is higher in ATP than in ATP+2APD, which is mentioned in the text, but this is inconsistent with the graph in 2E.

(4) The description of Results in the single-cell section is confusing, particularly in the sorted CD49 and unsorted cultures. Where do these cells come from? Are they BFP-, BFP+, unsorted for BFP, or non-patterned? Which are the "all three astrocyte populations"? A more complete description of the "iPSC-derived neurons" is required in this section to allow the reader to understand the type and maturation stage of neurons, and if they are patterned or not.

(5) A puzzling fact is that both BFP- and BFP- cells have similar levels of LMX1A, as shown in Fig. S6F. How do authors explain this observation?

(6) In Fig. 3B, the non-patterned cells cluster away from the BFP+ and BFP-; on the other hand, early and late BFP- are close and the same is true for early and late BFP+. A possible interpretation of these results is that patterned astrocytes have different paths for differentiation, compared to non-patterned cells. If that can be implied from these data, authors should discuss the alternative ways for astrocytes to differentiate.

(7) Fig. 3D shows that cluster 9 is the only one with detectable and coincident expression of both S100B and GFAP expression. Please discuss why these widely-accepted astrocyte transcripts are not found in the other astrocytes clusters. Also, Sox9 is expressed in neurons, astrocyte precursors and astrocytes. Why is that?

(8) Line 337, Why authors selected a log2 change of 0.25? Typically, 1 or a higher number is used to ensure at least a 2-fold increase, or a 50% decrease. A volcano plot generated by the comparison of BFP+ with BFP- cells would be appropriate. The validation of differences by immunocytochemistry, between BFP+ and BFP-, is inconclusive. The staining is blur in the images presented in Fig. S8C. Quantification of the positive cells, without significant background signal, in both populations is required.

(9) Lines 349-351: BFP+ cells did not show higher levels of transcripts for LMX1A nor FOXA2. This fact jeopardizes the claim that these cells are still patterned. In the same line, there are not significant differences with cortical astrocytes, indicating a wider repertoire of the initially patterned cells, that seems to lose the midbrain phenotype. Furthermore, common DGE shared by BFP- and BFP+ cells when compared to non-patterned cells indicate that after culture, the pre-pattern in BFP+ cells is somehow lost, and coincides with the progression of BFP- cells.

(10) For the GO analyses, How did authors select 1153 genes? The previous section mentioned 287 genes unique for BFP+ cells. The Results section should include a rationale for performing a wider search for the enriched processes.

(11) For Fig. 4C and 4D, both p values and the number of genes should be indicated in the graph. I would advise to select the 10 or 15 most significant categories, these panels are very difficult to read. Whereas the listed processes for BFP+ have a relation to Parkinson disease, the ones detected for BFP- cells are related to extracellular matrix and tissue development. Does it mean that BFP+ cells have impaired formation of this matrix, or defective tissue development? This is in contradiction of enhanced calcium responses of BFP+ cells compared to BFP- cells.

(12) Both the comparison between midbrain and cortical astrocytes in Fig. S8A, and the volcano plot in S8B do not show consistent changes. For example, RCAN2 in Fig. S8A has the same intensity for cortical and midbrain cells, but is marked as an enriched gene in midbrain in the p vs log2FC graph in Fig. S8B.

---

## [Author Response]

The following is the authors’ response to the current reviews.

**Response to Reviewer #3:**

We thank reviewer 3 for spending their valuable time on commenting on our revised paper.

We would like to reiterate the central conclusion of this work, which appears to have been missed by Reviewer 3. Using a BFP-expressing lineage tracer hPSC line for tracking LMX1A+ midbrain-patterned neural progenitors and their differentiated progeny, we discovered a loss of the LMX1A lineage during pluripotent stem cell differentiation into astrocytes, despite BFP+ neural progenitors were the dominant population at the onset of astrocyte induction.

Hence, the take-home message of this study is, as summarized in the abstract, ‘ the lineage composition of iPSC-derived astrocytes may not accurately recapitulate the founder progenitor population’ and that one should not take for granted that in vitro/stem cell-derived astrocytes are the descendants of the dominant starting neural progenitors (which is a general assumption in PSC publications as described in the paper and our response to reviewers).

Please find below our point-by-point response to reviewer comments. We have re-ordered the points according to their relative importance to our main conclusions.

‘ the lineage composition of iPSC-derived astrocytes may not accurately recapitulate the founder progenitor population’ and that one should not take for granted that in vitro/stem cell derived astrocytes are the descendants of the dominant starting neural progenitors (which is a general assumption in PSC publications as described in the paper and our response to reviewers).

Please find below our point-by-point response to their comments. We have re-ordered the points according to their relative importance to our main conclusions.

…. They used lineage tracing with a LMX1A-Cre/AAVS1-BFP iPSCs line, where the initial expression of LMX1A and Cre allows the long-lasting expression of BFP, yielding BFP+ and BFP- populations, that were sorted when in the astrocytic progenitor expansion. BFP+ showed significantly higher number of cells positive to NFIA and SOX9 than BFP- cells …

This is a misunderstanding by reviewer 3. As indicated in the first sentence of the second section, BFP- populations used for functional and transcriptomic analysis was not sorted BFP^-^ cells, but those derived from unsorted, BFP^+^ enriched populations. Our scRNAseq analysis indicated that they were transcriptomically aligned to human midbrain astrocytes. This finding is consistent with the fact that they are derived from midbrain-patterned neural progenitors, presumably minority LMX1A- progenitors.

Reviewer 3’s comments indicate that they misunderstood the primary aims of our study as a mere functional and transcriptomic comparison of the two astrocyte populations.

(9) BFP+ cells did not show higher levels of transcripts for LMX1A nor FOXA2. This fact jeopardizes the claim that these cells are still patterned. In the same line, there are not significant differences with cortical astrocytes, indicating a wider repertoire of the initially patterned cells, that seems to lose the midbrain phenotype. Furthermore, common DGE shared by BFP- and BFP+ cells when compared to non-patterned cells indicate that after culture, the pre-pattern in BFP+ cells is somehow lost, and coincides with the progression of BFP- cells.

The reviewer seems to assume that astrocytes derived from LMX1A+ ventral midbrain progenitors must retain LMX1A expression. We do not take this view and do not claim this in this study. Moreover, we have discussed in the paper that due to a lack of transcriptomic studies of in vivo track regional progenitors (such as LMX1A), it remains unknown whether and to what extent patterning gene expression is maintained in astrocytes of different brain regions.

Our findings on the lack of LMX1A and FOXA2 in BFP+ astrocytes are supported by several published single-cell transcriptomic studies of human midbrain astrocytes (La Manno et al. 2016; Agarwal et al. 2020; Kamath et al. 2022). We have a paragraph of discussion on this topic in both the original and updated versions of the paper with the relevant publications cited.

Other points raised by reviewer 3(1) It is very intriguing that GFAP is not expressed in late BFP- nor in BFP+ cultures, when authors designated them as mature astrocytes.

We did not designate our cells as ‘mature’ astrocytes but ‘astrocytes’ based on their global gene expression with the human fetal and adult brain astrocytes as references.

Moreover, ‘mature’ only appeared once in the paper indicating that our cells lie in between the fetal and adult astrocytes in maturity.

(2) In Fig. 2D, authors need to change the designation "% of positive nuclei".To be corrected in the version of record.(3) In Fig. 2E, the text describes a decrease caused by 2APB on the rise elicited by ATP, but the graph shows an increase with ATP+2APB. However, in Fig. 2F, the peak amplitude for BFP+ cells is higher in ATP than in ATP+2APD, which is mentioned in the text, but this is inconsistent with the graph in 2E.

To be corrected in the version of record.

(4) The description of Results in the single-cell section is confusing, particularly in the sorted CD49 and unsorted cultures. Where do these cells come from? Are they BFP-, BFP+, unsorted for BFP, or non-patterned? Which are the "all three astrocyte populations"? A more complete description of the "iPSC-derived neurons" is required in this section to allow the reader to understand the type and maturation stage of neurons, and if they are patterned or not.

As previously reported in the reference cited, CD49 is a novel human astrocyte marker. This is independent of BFP expression. For all three astrocyte populations studied here (BFP+, BFP-, and non-patterned astrocytes), we included both CD49f+ sorted and unsorted samples to account for selection bias caused by FACS. iPSC-derived neurons were included in the sequencing study to provide a reference for cell-type annotation. They were generated following a GABAergic neuron differentiation protocol. However, their maturation stages and/or regional characteristics are not relevant to astrocytes.

(5) A puzzling fact is that both BFP- and BFP- cells have similar levels of LMX1A, as shown in Fig. S6F. How do authors explain this observation?

This figure panel shows that LMX1A, LMX1B and FOXA2 are essentially NOT expressed in these astrocytes.

(6) In Fig. 3B, the non-patterned cells cluster away from the BFP+ and BFP-; on the other hand, early and late BFP- are close and the same is true for early and late BFP+. A possible interpretation of these results is that patterned astrocytes have different paths for differentiation, compared to non-patterned cells. If that can be implied from these data, authors should discuss the alternative ways for astrocytes to differentiate.

Both BFP+ and BFP- astrocyte are from ventral midbrain patterned neural progenitors, while non-patterned neural progenitors are more akin to that of forebrain. Figure 3B is expected and confirms the patterning effect.

(7) Fig. 3D shows that cluster 9 is the only one with detectable and coincident expression of both S100B and GFAP expression. Please discuss why these widely-accepted astrocyte transcripts are not found in the other astrocytes clusters. Also, Sox9 is expressed in neurons, astrocyte precursors and astrocytes. Why is that?

S100B and GFAP are classic astrocyte markers in certain states. We are not relying only on two markers but the genome-wide expression profile as the criteria for astrocytes. As shown in the unbiased reference mapping to multiple human brain astrocyte scRNA-seq datasets, all our astrocyte clusters were mapped with high confidence to human astrocytes.

SOX9 is an important regulator for astrogenesis, so its expression is expected in precursors (doi.org/10.1016/j.neuron.2012.01.024). In addition, recent studies have uncovered that SOX9 expression is also reported in foetal striatal projection neurons and early postnatal cortical neurons, where SOX9 regulates neuronal synaptogenesis and morphogenesis (dois:10.1016/j.fmre.2024.02.019; 10.1016/j.neuron.2018.10.008). Therefore, the expression of SOX9 in multiple cell types was expected. Instead of using a few selected markers for cell-type annotation, we employed a genomic approach relying on an unbiased reference mapping approach and a combination of various markers to ascertain our annotation results.

(8) Line 337, Why authors selected a log2 change of 0.25? Typically, 1 or a higher number is used to ensure at least a 2-fold increase, or a 50% decrease. A volcano plot generated by the comparison of BFP+ with BFP- cells would be appropriate. The validation of differences by immunocytochemistry, between BFP+ and BFP-, is inconclusive. The staining is blur in the images presented in Fig. S8C. Quantification of the positive cells, without significant background signal, in both populations is required.

We used a lenient threshold owing to the following considerations: (1) High FC does not necessarily mean biological relevance, as gene expression does not necessarily translate to protein expression. Therefore, a smaller FC value could also be biologically meaningful. (2) Balance between noise and biological differences. Any threshold was chosen arbitrarily. (3) We are identifying a trend rather than pinpointing a specific set of

The quality was unfortunately reduced due to restrictions on file size upon submission. A high resolution Fig. S8C is available.

(10) For the GO analyses, How did authors select 1153 genes? The previous section mentioned 287 genes unique for BFP+ cells. The Results section should include a rationale for performing a wider search for the enriched processes.

GO enrichment using unique DEGS may not capture the wider landscape of the transcriptomic characteristics of BFP^+^ astrocytes. The 287 unique genes were only differentially expressed in BFP^+^ astrocytes. However, apart from these 287 genes, other genes among the 1187 DEGs were differentially expressed in BFP^+^ astrocytes and in one other population.

(11) For Fig. 4C and 4D, both p values and the number of genes should be indicated in the graph. I would advise to select the 10 or 15 most significant categories, these panels are very difficult to read. Whereas the listed processes for BFP+ have a relation to Parkinson disease, the ones detected for BFP- cells are related to extracellular matrix and tissue development. Does it mean that BFP+ cells have impaired formation of this matrix, or defective tissue development? This is in contradiction of enhanced calcium responses of BFP+ cells compared to BFP- cells.

Information on all DEGs, including p values and numbers, is provided in Supplementary data 1-5.

BFP+ astrocytes do have enrichment for GO terms related to extracellular matrix and tissue development, although not as obvious as BFP- astrocytes. Previous work have shown that both in vitro and in vivo derived astrocytes are functionally heterogeneous, containing functionally distinct subtypes exhibiting different GO enrichment profiles (doi: 10.1016/j.ygeno.2021.01.008; 10.1038/s41598-024-74732-7).

(12) Both the comparison between midbrain and cortical astrocytes in Fig. S8A, and the volcano plot in S8B do not show consistent changes. For example, RCAN2 in Fig. S8A has the same intensity for cortical and midbrain cells, but is marked as an enriched gene in midbrain in the p vs log2FC graph in Fig. S8B.

These are integrated analyses of published human datasets. S8A and S8B show the same data in different formats. The differences are better shown in the volcano plot/easier detected by the human eye.

These are integrated analysis of published human datasets. S8A and S8B are the same data shown in different format. Differences are better shown in volcano plot /easier detected by the human eye. RCAN2 had a higher average expression in the midbrain than in the telencephalon, albeit small, and the difference was statistically significant (as shown in the volcano plot).

The following is the authors’ response to the original reviews

**Reviewer 1:**
In vitro nature of this work being the fundamental weakness of this paper

We disagree with this statement. As explained in the provisional response, the aim of this study was to test the validity of a general concept applied in pluripotent stem cell research that pluripotent stem cell-derived astrocytes faithfully represent the lineage heterogeneity of their ancestral neural progenitors and hence preserve the regionality of such progenitors. Our genetic lineage study is justified for addressing this in vitro-driven question. However, we have highlighted the rationale where appropriate in the revised paper.

If regional identity is not maintained, so what? Don't we already know that this can happen? The authors acknowledge that this is known in the discussion.

Importance of regional identity: Growing evidence demonstrates the functional heterogeneity of brain astrocytes in health and disease. Therefore, for in vitro disease modeling, it is believed that one should use astrocytes represent the anatomy of disease pathology; for example, midbrain astrocytes for studying dopamine neurodegeneration and Parkinson’s disease. Understanding the dynamics of stem cell-derived astrocytes and identifying astrocyte subtypes is important for their biomedical applications.

Regional identity change/Discussion: It seems that the reviewer misunderstood the context in which the ‘identity change’ was discussed. The literature referred to (in the Discussion) concerns shifts in regional gene expression in bulk-cultured cells. In the days of pre-single-cell analysis/lineage tracking, one cannot distinguish whether this was due to a change in the transcriptomic landscape in progenies of the same lineage or alterations in lineage heterogeneity, but to interpret at face value as regional identity was not maintained. In the revised paper, we have made an effort to indicate that ‘regional identity’ is used broadly to refer to lineage relationships and/or traits rather than static gene expressioin.

validation of the markers/additional work

The scNAseq analysis performed in this study compared the profiles of astrocytes derived from LMX1A+ and LMX1A- ventral midbrain-patterned neural progenitors. Since it is not possible to perform genetic lineage tracking in humans and an analogous mouse lineage tracer line is not available, in vivo validation of these markers with respect to their lineage relationship is not currently feasible. However, we took advantage of abundant single-cell human astrocyte transcriptomic datasets and validated our genes in silico. We also validated the differential expression of selected markers in late BFP+ and BFP- astrocytes using immunocytochemistry, where reliable antibodies are available. The results of the additional analyses are presented in Figure S8 and Supplemental Data 5.

Knowledge gaps concerning astrocyte development

Reviewer 1 pointed out a number of knowledge gaps concerning astrocyte development, such as the transcriptomic landscape trajectories of midbrain floor plate cells as they progress towards astrocytes. Indeed, the limited knowledge on regional astrocyte molecule heterogeneity restricts the objective validation of in vitro-derived astrocyte subtypes and the development of novel approaches for their generation in vitro. We agree with the need for in-depth in vivo studies using model organisms, although these are beyond the scope of the current work.

**Reviewer 2:**
(1) The authors argue that the depletion of BFP seen in the unsorted population immediately after the onset of astrogenic induction is due to the growth advantage of the derivatives of the residual LMX1A- population. However, no objective data supporting this idea is provided, and one could also hypothesize that the residual LMX1A- cells could affect the overall LMX1A expression in the culture through negative paracrine regulation.

We acknowledge the lack of evidence-based explanation for the depletion of BFP+ cells in mixed cultures. We were unable to perform additional experiments because of resource limitations. The design of the LMX1A-Cre/AAVS1-BFP lineage tracer line determines that BFP is expressed irreversibly in LMX1A-expressing cells or their derivatives regardless of their LMX1A expression status. Therefore, the potential negative paracrine regulation of LMX1A by residual LMX1A- cells should not affect cells that have already turned on BFP. We have highlighted the working principles of the LMX1A tracer line in the revised manuscript.

(2) Furthermore, on line 124 it is stated that: "Interestingly, the sorted BFP+ cells exhibited similar population growth rate to that of unsorted cultures...". In the face of the suggested growth disadvantage of those cells, this statement needs clarification.

To avoid confusion, we have removed the statement.

(3) Regarding the fidelity of the model system, it is not clear to me how the TagBFP expression was detected in the BFP+ population supposedly in d87 and d136 pooled astrocytes (Fig S6C) while no LMX1A expression was observed in the same cells (Fig S6F).

The TagBFP tracer is expressed in the progenies of LMX1A+ cells, regardless of their LMX1A expression status. We have gone through the MS text to ensure that this information has been provided.

(4) The generated single-cell RNASeq dataset is extremely valuable. However, given the number of conditions included in this study (i.e. early vs late astrocytes, BFP+ vs BFP-, sorted vs unsorted, plus non-patterned and neuronal samples) the resulting analysis lacks detail. For instance, from a developmental perspective and to better grasp the functional significance of astrocytic heterogeneity, it would be interesting to map the identified clusters to early vs late populations and to the BFP status.

We performed additional bioinformatics analysis, which provided independent support for the relative developmental maturity suggested by functional assays. The additional data are now provided in the revised Figure 3B, C, E.

Moreover, although comprehensive, Figure S7 is complex to understand given that citations rather than the reference populations are depicted.

The information provided in the revised Figure S7.

(5) Do the authors have any consideration regarding the morphology of the astrocytes obtained in this study? None of the late astrocyte images depict a prototypical stellate morphology, which is reported in many other studies involving the generation of iPSC-derived astrocytes and which is associated with the maturity status of the cell.

The morphology of our astrocytes was not unique to the present study. Many factors may influence the morphology of astrocytes, such as the culture media and supplements used, and maturity status. Based on the functional assays and limited GFAP expression, our astrocytes were relatively immature.